# Reactive Oxygen Species: A Crosslink between Plant and Human Eukaryotic Cell Systems

**DOI:** 10.3390/ijms241713052

**Published:** 2023-08-22

**Authors:** Wei Guo, Yadi Xing, Xiumei Luo, Fuguang Li, Maozhi Ren, Yiming Liang

**Affiliations:** 1Zhengzhou Research Base, National Key Laboratory of Cotton Bio-Breeding and Integrated Utilization, School of Agricultural Sciences, Zhengzhou University, Zhengzhou 450001, China; gw354518@gmail.com (W.G.); xingyadi@zzu.edu.cn (Y.X.); aylifug@caas.cn (F.L.); 2National Key Laboratory of Cotton Bio-Breeding and Integrated Utilization, Institute of Cotton Research, Chinese Academy of Agricultural Sciences, Anyang 455000, China; 3Institute of Urban Agriculture, Chinese Academy of Agricultural Sciences, Chengdu 610000, China; luoxiumei@caas.cn; 4Hainan Yazhou Bay Seed Laboratory, Sanya 572000, China

**Keywords:** reactive oxygen species (ROS), regulatory factors, oxidative stress, cellular response, ROS scavenging

## Abstract

Reactive oxygen species (ROS) are important regulating factors that play a dual role in plant and human cells. As the first messenger response in organisms, ROS coordinate signals in growth, development, and metabolic activity pathways. They also can act as an alarm mechanism, triggering cellular responses to harmful stimuli. However, excess ROS cause oxidative stress-related damage and oxidize organic substances, leading to cellular malfunctions. This review summarizes the current research status and mechanisms of ROS in plant and human eukaryotic cells, highlighting the differences and similarities between the two and elucidating their interactions with other reactive substances and ROS. Based on the similar regulatory and metabolic ROS pathways in the two kingdoms, this review proposes future developments that can provide opportunities to develop novel strategies for treating human diseases or creating greater agricultural value.

## 1. Introduction

Reactive oxygen species (ROS) include various forms such as O_2_^•−^, H_2_O_2_, OH^•^, ^1^O_2_, and so forth [1], which participate in the development, growth, differentiation, and proliferation of multicellular organisms. Different types of reactive oxygen species (ROS) possess varying half-lives and affinities for biological molecules. For instance, ^1^O_2_ and OH^•^ are highly reactive and short-lived species, while H_2_O_2_ exhibits lower reactivity and a longer half-life. This characteristic makes H_2_O_2_ more suitable as a signaling molecule, whereas the former examples are known to signal through their breakdown products. Moreover, the production dynamics and subcellular localization of ROS can vary depending on the physiological state of the cell. Chloroplasts can produce both ^1^O_2_ and O_2_^•−^ when the balance between light reactions and Calvin reactions is disrupted, typically caused by excessive excitation of photosystems or insufficient CO_2_ supply. Mitochondria, on the other hand, can generate O_2_^•−^ when the electron transport chain becomes overloaded. H_2_O_2_ production, through the dismutation of O_2_^•−^, can occur in all cellular compartments [2]. In animals, ROS are often associated with a wide variety of pathologies, such as cancer, neurodegeneration, atherosclerosis, diabetes, and aging [3]. However, recent research has revealed that ROS can also be involved in metabolic regulation and stress responses to support cellular adaptation to changing environments and stresses [4]. In contrast, ROS seem to be relatively less harmful to plant cells, and research has focused on the regulation systems in which ROS are involved [5], including the cell cycle, abiotic stress response, systemic signaling, programmed cell death, pathogen defense, and development. Nonetheless, excessive ROS accumulation can lead to lipid peroxidation, membrane disintegration, and damage to DNA, proteins, and carbohydrates [6]. The purpose of this review is to summarize the study’s findings by comparing the similarities and differences in how plants and animals use ROS to regulate relevant physiological responses and the role of ROS in cellular damage and repair.

## 2. ROS in Plants

### 2.1. Cell Cycle

Plants rely on ROS to regulate their cell cycle, which is defined as the set of events through which a cell grows, replicates its genome, and ultimately divides into two daughter cells via mitosis. This complex process is controlled by various signaling pathways and regulatory proteins. Studies have shown that sustained mild oxidation can impair cell cycle progression in *Arabidopsis*, leading to a reduction in cells in root tip meristematic tissue [7]. In *Chlamydomonas reinhardtii*, the application of exogenous H_2_O_2_ alters cell cycle processes, and the timing of key Chlamydomonas cell cycle checkpoints correspond with changes in the concentrations of H_2_O_2_, such as G1/S and S/M transitions or progeny cell release [8,9].

ROS affect the cell cycle by regulating cyclin-dependent kinases (CDKs), which play a central role in cell cycle regulation. Depending on the specific type and concentration of ROS, CDKs can be activated or inhibited, thereby affecting cell cycle progression. High levels of ROS, for instance, can inhibit CDK activity, leading to a delay in the G1 phase of the cell cycle. Conversely, low levels of ROS can stimulate CDK activity and promote cell cycle progression (Figure 1B). In maize, Tyr nitration in CDKA;1 can act as an active regulator of cell cycle progression during redox stress and is responsible for regulating cell proliferation during plant stress [10]. In wheat, oxidative treatment resulted in reduced CDKD and CDKA protein ubiquitination, leading to decreased cellular CDKD/CDKA/Rb/E2F regulatory gene expression and affecting G1/S transition and S-phase progression [11]. Additionally, the activation of the plant CDK complex to promote cell entry into the G0/G1 phase was enhanced by the synergistic effect of ROS and growth hormone [12].

Aside from affecting individual proteins, ROS can alter the overall balance of signaling pathways that regulate the cell cycle. For example, ROS can regulate kinase activity to affect cell cycle progression. Capsicum annuum receptor-like kinase 1 (CaRLK1) promotes the transition from G0/G1 phase to S phase by controlling O_2_^•−^ levels while inducing catalase (CAT) activity and protein peroxidase. This, in turn, reduces the level of H_2_O_2_ [13]. CRK6 and CRK7 have been found to assist plants in coping with elevated ROS levels in the plastid caused by O_3_, ensuring that the cell cycle progresses normally [14]. 

Another way in which ROS regulates the cell cycle is through the influence of transcription factors. Transcription factors are proteins that bind to DNA and regulate gene expression [15]. ROS can modify the activity of transcription factors by altering their structure or the DNA to which they bind [16]. This modification can result in changes in the expression of genes that are essential for cell cycle regulation, such as cell cycle proteins and CDKs. Bursts of ROS can activate the MYB30 transcription factor network, which is involved in regulating the balance between root growth and defense [17]. *Arabidopsis* prohibitin protein (PHB3) is also involved in regulating ROS homeostasis and, thus, controls root meristem size and root stem cell niche (SCN) maintenance through the ROS response ethylene response factor (ERF) transcription factors ERF109, ERF114, and ERF115 that have been identified [18].

The relationship between ROS and the plant cell cycle is complex and multifaceted. While ROS can have both positive and negative effects on the cell cycle, it is evident that they play a crucial role in regulating plant cell growth and division.

### 2.2. Plant Growth

ROS have both positive and negative effects on plant growth and development. On the positive side, ROS have been shown to stimulate cell division and expansion and promote plant growth and development under normal conditions. However, excessive ROS production can lead to oxidative stress, causing damage to plant cells and leading to plant tissue death (Figure 1C).

During the process of seeds uptaking water, the subcellular compartmentalization of ROS and its target molecules regulates the expression of various genes [19]. Furthermore, many unstable H_2_O_2_ messengers depend on the redox state of active proteins to accelerate redox-sensitive transcription factors, thereby activating a downstream cascade that triggers the oxidation of MAPKs and specific peptides. This oxygenated protein inhibits translation by oxidizing mRNA while driving germination [20,21]. A seed-specific peroxiredoxin AtPER1 eliminates ROS to inhibit abscisic acid (ABA) catabolism and gibberellin (GA) biosynthesis, thereby ameliorating primary seed dormancy and making seeds less sensitive to adverse environmental conditions [22]. 

In lateral root development, ROS can promote the transition from cell proliferation to differentiation in lateral roots [23]. At the same time, ROS can activate the MAPK cascade to promote the development of root hairs [24]. In this process, flavanols can regulate lateral root emergence and root hair development by scavenging reactive oxygen species in *Arabidopsis* [25,26]. 

In plant SAM (shoot apical meristem), ROS can trigger the phase separation of TMF (terminating flower) to direct stem cell fate for the flowering transition [27]. The rapid alkalinization factor peptides RALF23 and RALF33 are known to induce ROS production in stigma papillae. Meanwhile, post-pollination pollen-coating protein class B peptides (PCP-Bs) compete with RALF23/33 to bind ANJEA (ANJ)-FERONIA (FER) complexes, reducing stigma ROS and promoting pollen hydration and germination [28]. FPF1-like protein 4 (OsFPFL4) can be involved in root and flower development by affecting growth hormone levels and ROS accumulation in rice [29].

ROS plays a significant role in plant fertilization. Upon landing on the stigma, pollen grains germinate and give rise to pollen tubes, which grow towards the ovules inside the flower. ROS, particularly hydrogen peroxide (H_2_O_2_), are generated in the growing pollen tubes. H_2_O_2_ acts as a signaling molecule that regulates the elongation of pollen tubes. It is involved in various processes, including reorientation of cytoskeletal components, cell wall modifications, and regulation of ion channels, all of which are crucial for successful pollen tube growth and guidance towards the ovules. Once the pollen tubes reach the ovules, they encounter specific structures such as the embryo sac or receptive synergid cells. These structures produce ROS, including superoxide radicals (O_2_^•−^) and H_2_O_2_. ROS participate in crosstalk between the pollen tube and the receiving cells, facilitating successful fertilization. ROS contribute to guiding the pollen tubes towards the ovules and mediate cellular processes that promote pollen tube reception and subsequent fusion with the egg cell [30].

Leaf cell expansion requires changes in the structure and content of plant cell walls, and these cell wall changes are mediated by peroxidase-associated ROS gradients. Plastid extracellular peroxidases directly regulate cell wall rigidity by limiting or promoting cell extension [31]. A transcription factor, KUODA1 (KUA1), specifically controls cell expansion during *Arabidopsis* leaf development. KUA1 directly represses the expression of a set of genes encoding peroxidases, and disruption of KUA1 results in increased peroxidase activity and smaller leaf cells [32]. The chloroplast-localized NAD kinase (NADK2) mutant *Arabidopsis* produced higher levels of ROS, exhibited delayed growth under continuous light conditions, and showed more severe growth inhibition under short daylight [33].

### 2.3. Abiotic Stress Response

Drought stress and stress caused by high soil salinity are the two most common sources of abiotic plant stress, and there is now a wealth of relevant evidence for the involvement of ROS in the key regulation of abiotic stress responses (Figure 1B) [34,35,36,37]. The first is the accumulation of ROS due to drought. ROS accumulation due to drought induces stomatal closure in response to downstream ABA through a complex signaling network [38,39]. Subsequently, to avoid cell damage due to ROS accumulation, plants up-regulate the activity of a series of peroxidases and their regulators (NADPH oxidases, heterotrimeric guanine-nucleotide-binding proteins (G-proteins), etc.) to maintain the dynamic balance of ROS [40,41]. 

In salt stress, ROS can stimulate different signaling molecules or hormones that have diverse physiological, growth, and developmental functions in the salt adaptation of plants. For instance, in rice, the salt-induced protein kinase SIT1 is rapidly activated under salt stress, leading to the activation of MAPKs such as MPK3/6, which promotes ethylene synthesis and ROS production. These processes improve salt tolerance in rice by enhancing ion homeostasis, reducing oxidative damage, and regulating gene expression to maintain plant growth and development under salt stress [42]. AtrbohF is a NADPH oxidase that plays a role in mediating ROS production in response to salt stress. The increased ROS levels in the root vascular tissue can limit the accumulation of sodium ions (Na^+^) in the xylem sap, which in turn reduces Na^+^ transport from roots to branches. This results in a lower Na^+^/K^+^ ratio and contributes to soil salinity tolerance in plants [43]. Genes such as OsMADS25 in rice and GhWRKY17 in cotton can regulate peroxidase activity under salt stress, while also controlling stomatal opening frequency through the ABA signaling pathway [44,45]. Recent studies have shown that some miRNAs can also be involved in the dynamic regulation of ROS to enhance salt tolerance. For example, the miR172/IDS1 signaling module in cereals and the ZmmiR169q/ZmNF-YA8 module in maize maintain ROS homeostasis by controlling peroxidase activity [34,37].

Under high-temperature stress, Ca^2+^ can activate RBOH proteins, leading to a burst of ROS in the plasmodesmata. The generated ROS are then transported into the cell via water channel proteins, triggering further release of Ca^2+^ from TPC1 channels. This process helps regulate different signaling and stress responses [46,47]. Genes regulating ROS are present in brassinosteroids (BRs), absiaisic acid (ABA), and jasmonate (JA) signaling pathways, coordinating appropriate molecular and physiological adaptive responses to high temperature. Furthermore, there is ample evidence that ROS are involved in the activation of heat shock proteins in the early stages of heat stress, allowing plants to acquire tolerance [48,49]. Similarly, under low temperature stress Ca^2+^ can also lead to activation of RBOH and thus produce ROS to regulate downstream responses (Figure 1D) [50]; the signaling pathways of BRs, ABA, and ET can also dynamically regulate ROS [51,52,53,54]. 

To alleviate oxidative stress induced by various abiotic stresses, other reactive oxygen signaling molecules also participate in the regulatory processes. Hydrogen sulfide (H_2_S) is one such molecule that regulates the synthesis of glutathione (GSH) in plants. It helps in the removal of excessive reactive oxygen species (ROS) through the ascorbate-glutathione pathway [55]. Under drought conditions, hydrogen sulfide (H_2_S) can participate in the sulfhydration of certain proteins, thereby synergistically mediating stomatal closure along with ROS [56].

### 2.4. Programmed Cell Death

Programmed cell death (PCD) is common during the development of organisms, and ROS play an important role in this process (Figure 1C) [57,58,59]. Mitochondrial morphological transition is an early and specific indicator of cell death and a necessary component of the cell death process. There is evidence that elevated levels of Ca^2+^ in PCD can accelerate mitochondrial morphological transition, leading to increased levels of ROS [60,61]. ROS also act as signaling molecules that activate the permeability transition channel of the mitochondrial membrane. This leads to the rapid release of cytochrome *c* from mitochondria, disrupting the electron transport chain and resulting in high levels of ROS production. These high levels of ROS production can ultimately lead to irreversible PCD development [62]. In stem cells, the exogenous application of H_2_O_2_ inhibits the expression of the stem cell determinant *WUSCHEL (WUS)* and promotes the expression of the developmental programmed cell death (dPCD) marker gene *ORESARA 1 (ORE1)* [63]. In *Arabidopsis*, methionine sulfoxide reductase 8 (MSRB8) can regulate the levels of ROS, activate effector-triggered immunity (ETI), and suppress stress-induced cell death [64].

In plants, the malate shuttle may provide direct communication signals from chloroplasts to mitochondria via cytoplasmic lysates, delivering reducing equivalents for mitochondrial ROS production and inducing ROS production [65]. Light-induced ROS can lead to fumonisin B1 (FB1)-induced degradation of chloroplast proteins and act as a second messenger to enhance phenylalanine ammonia lyase (PAL) activity, inducing PCD by upregulating salicylic acid synthesis [66]. The chloroplast is clearly involved in the amplification of cellular ROS and the cellular redox changes effected by it could act as a signaling event that depending on the physiological state of the cell could lead to PCD. Based on the available information discussed above, the chloroplasts act as one of the prominent executioners of the PCD process in plants [67]. 

### 2.5. Pathogen Defense

During the initial stages of pathogen infection, an oxidative burst is necessary to activate local and systemic defense responses (Figure 1E). The early recognition response to pathogen attack aims to confine pathogen propagules to their entry sites through on-site programmed cell death (PCD), resulting in a phenomenon known as “hypersensitivity”. In the later stages of infection, when the hypersensitivity response inhibits pathogen infection, it becomes necessary to neutralize the excess ROS by initiating antioxidant defense mechanisms [68]. In certain situations, reactive oxygen species (ROS) can engage in synergistic reactions with other reactive oxygen species to combat pathogen invasion. Nitric oxide (NO) can regulate cell death in plants by *S*-nitrosylating NADPH oxidase [69]. In the meantime, numerous proteins are both NO and H_2_O_2_ targets. For instance, H_2_O_2_ directly targets GAPDH, which is involved in mediating ROS signaling in plants, and NO-mediated *S*-nitrosylation targets it as well, which reduces glyceraldehyde-3-phosphate dehydrogenase’s (GAPDH) activity [70]. Melatonin has been reported to trigger downstream MAPK cascades, such as MAPK3 and MAPK6, in response to a ROS burst, thereby mediating the expression of defense-related genes [71]. Melatonin can also inhibit the production of mitochondrial peroxides in plants and participate in regulating ROS levels and maintaining redox homeostasis in plant mitochondria, thereby preventing the damage caused by excessive ROS [72]. The redox sensor QSOX1 is a powerful tool for targeting the redox level in plants, regulating plant immunity by targeting S-nitrosoglutathione reductase (GSNOR) to regulate ROS production [73]. Water channel proteins, such as plasma membrane intrinsic proteins (PIPs) and tonoplast intrinsic proteins (TIPs), known as aquaporins (AQPs) in plants, mediate the transport of H_2_O_2_ across cell membranes, thereby triggering a variety of immune responses, such as PTI (PAMP-triggered immunity) and SAR (Systemic acquired resistance) [74]. Some plant volatile organic compounds (VOCs) such as monoterpenes can act through ROS to promote SAR [75]. Polyamines (PAs) are involved in increasing H_2_O_2_ levels through the catabolism of amine oxidase, and advanced PAs, such as spermine, can also act as hydroxyl radical scavengers [76]. The primary distinguishing factor between plant and human cells is the presence of the plant cell wall. This wall serves as a crucial physical barrier that pathogens must navigate when colonizing plant tissues. Disruptions to cell wall integrity can trigger targeted defense responses by affecting proteins engaged in cell wall biosynthesis, degradation, and remodeling. Such disruptions can arise from cell wall damage caused by both biotic and abiotic stresses. Plants employ inducible defense mechanisms that involve biochemical alterations of cell wall components, shielding them from enzymatic deterioration. Notably, ROS play a pivotal role in orchestrating cell wall remodeling in reaction to pathogens and plant viruses [77,78,79,80]. ROS can act as antimicrobial molecules involved in reinforcing plant cell walls and inducing callus deposition to limit pathogen entry [81]. The enzymatic activities of peroxidases [82] and pectin methylesterases [83], influenced by ROS, contribute to the restructuring of cell wall polymers, offering a dynamic response to the evolving threat. In the Arabidopsis–TuMV pathosystem response, glutathione can act as a key signaling factor in not only the susceptible *rbohD* reaction but also the resistance reaction presented by *rbohF* and *rbohD/F* mutants [84]. Another study demonstrated that proline dehydrogenase in *Arabidopsis* promotes flagellin-mediated PAMP-triggered immune responses by affecting RBOHD [85]. Interestingly, symbiotic plants have evolved a ROS-inhibitory response to lipochitin oligosaccharides (LCOs) to facilitate the early steps of symbiosis while also maintaining a parallel defense mechanism against pathogens [86]. After nematode invasion in plants, plants undergo oxidative bursts to resist nematode invasion. In *Arabidopsis*, ten homologs of RBOHs (Respiratory burst oxidase homologs) are elevated during nematode invasion [87]. Simultaneously, nematodes secrete ROS scavengers, such as SOD (superoxide dismutase), catalase, thioredoxins, and glutathione peroxidase, to suppress plant immune responses [88]. Additionally, some nematode-secreted effectors target the plant’s ROS system, making it more susceptible to nematode invasion [89,90,91].

### 2.6. Aging

Reactive oxygen species (ROS) are highly reactive molecules that are byproducts of normal cellular processes, such as respiration and photosynthesis. These molecules can cause damage to cellular components, including DNA, proteins, and lipids, leading to plant senescence. Plants have developed several mechanisms to protect themselves from ROS, including the production of antioxidant enzymes such as superoxide dismutase and catalase. These enzymes help neutralize ROS and prevent them from causing damage (Figure 1B). In rice, Leaf senescence 1 (LS1) may regulate leaf development and function. *LS1* mutants exhibit premature leaf senescence and reduced chlorophyll content, with significant increases in ROS and SOD content and significant reductions in CAT activity [92]. WRKY75, SA, and ROS could form three interrelated positive feedback loops to amplify accelerated leaf senescence [93].

Plants approaching senescence significantly reduce the rate of producing these protective enzymes, potentially making them more susceptible to ROS-induced damage. This may lead to the development of age-related diseases and ultimately plant death [94].

In addition, environmental factors such as strong light, drought, and insect pests can increase the production of reactive oxygen species in plants and accelerate the aging process. Chronic doses of ozone can reduce photosynthesis, destroy mitochondria, and accelerate the programmed cell death process in cells [95]. During drought-induced leaf senescence, the drought-responsive NAC transcription factor NTL4 promotes the production of ROS by directly binding to the promoter of the gene encoding the ROS biosynthetic enzyme [96]. Under salt stress, AtCaN2 leads to the accumulation of excess H_2_O_2_ and accelerates plant senescence [97]. However, there are some molecules that can delay plant senescence and enable them to extend their lifespan. For example, CLE14 can suppress age-dependent and stress-induced leaf senescence by promoting JUB1-mediated ROS clearance in *Arabidopsis* [98]. Melatonin MT inhibits age-dependent and stress-induced leaf senescence in cucumber by activating the antioxidant system and the IAA synthesis and signaling pathway, while simultaneously inhibiting the ABA synthesis and signaling pathways, thereby reducing leaf senescence [99]. The miR775-GALT9 module regulates leaf senescence in *Arabidopsis* through crosstalk between the ethylene signaling and ABA biosynthetic pathways during post-submergence recovery [100].

### 2.7. System Signaling

As mentioned above, ROS usually appear as first or second messengers when acting as signaling molecules, which is determined by the molecular properties of ROS. Water-soluble small molecules of ROS can rapidly generate bursts via oxidative enzymes such as respiratory burst oxidase homologs (RBOHs) or external factors such as UV light and then efficiently conduct signals through the vascular tissues of plants while entering cells to act with the help of water channel proteins. These ROS signals are considered as ROS waves and, in *Arabidopsis*, they can be transmitted to the whole plant in only 20–30 min, which can alert cells and tissues to impending stress and accompany other signals that may convey specificity [101]. Two different pathways have been reported under light stimulation to mediate the rapid transmission of systemic ROS signals from their origin to the whole plant (Figure 1E). 

The first is a cell-autonomous pathway that amplifies systemic ROS signals, triggers acclimation responses, and requires RBOHD, GLR3.3, GLR3.6, PIP2;1, CNGC2, and MSL2. The second is a cell-to-cell pathway that propagates the systemic signal and requires RBOHD, PDLP5, and PDLP1. This evidence suggests that ROS produced by RBOH can be processed via PDLPs to enhance intercellular transport and PD pore size, a mechanism that can control the mobilization of many different systemic signals in plants [102]. 

Another study showed that in pathogen defense, CPK5 can amplify ROS signals in a similar way by phosphorylating RBOHD and cascading ROS signals [103]. ROS waves can link abiotic and biotic signals during SAR, soil water repellency (SWR), or systemic acquired acclimation (SAA). When biotic and abiotic conditions or stresses are combined in abiotic–biotic stress combinations, the plant response is unique compared to each stress applied alone [101].

### 2.8. ROS Hazard and Scavenging

In plants, photosynthesis relies on the light-dependent reactions, where plants utilize chlorophyll and other pigments to capture light energy. This energy is then used to drive a series of redox reactions in the electron transport chain (ETC) located in the thylakoid membrane of chloroplasts. During this process, water molecules are split, releasing electrons and protons. The flow of electrons through protein complexes in the ETC ultimately reduces NADP^+^ (nicotinamide adenine dinucleotide phosphate) to NADPH, a molecule with high reducing power. This process generates a proton gradient across the thylakoid membrane, which is used by ATP synthase to produce ATP. This proton gradient is also one of the main sources of ROS production in plants [104]. In the Calvin cycle, which takes place in the stroma of chloroplasts, plants utilize ATP and NADPH generated from the light-dependent reactions to convert carbon dioxide into glucose and other carbohydrates. The Calvin cycle involves a series of redox reactions, including the reduction of carbon dioxide and the regeneration of the initial molecule, 1,5-bisphosphoribulose (RuBP). Ribulose-1,5-bisphosphate carboxylase/oxygenase (Rubisco) plays a central role in this process. The redox balance is maintained through the interaction of NADPH, NADP^+^, and other redox carriers within the chloroplast [105]. In cellular respiration, which occurs in the mitochondria, organic molecules such as glucose are broken down to release energy. This process involves a series of redox reactions in the mitochondrial electron transport chain (ETC). Electrons derived from the breakdown of organic molecules are transferred to electron carriers such as NADH (reduced form of nicotinamide adenine dinucleotide) and FADH2 (reduced form of flavin adenine dinucleotide), which provide electrons to the ETC. The flow of electrons through protein complexes in the ETC leads to ATP production through oxidative phosphorylation. Oxygen serves as the final electron acceptor, forming water. The redox balance is maintained through the transfer of electrons and protons within the ETC [106]. In addition to the redox reactions occurring in photosynthesis and cellular respiration, the cellular redox state also influences the activity and regulation of many enzymes and signaling pathways in plants. Redox signaling pathways mediated by hydrogen peroxide (H_2_O_2_) are typically involved in various plant responses, including stress adaptation, growth, and development [107]. Plants dynamically regulate ROS levels through a variety of crosstalk reactions, as excess ROS can be harmful to them (Figure 1A). Post-translational modifications (PTMs) of catalase can be collectively regulated by NO and H_2_S, including *S*-nitrosylation, tyrosine nitration (Tyr-NO), metal nitrosylation, and sulfhydration. These modifications collectively regulate the activity of catalase [108]. 

The various molecular forms of ROS have different roles and functions. Superoxide radical (O_2_^•−^) is an important reactive oxygen species (ROS) molecule in plants. It is a byproduct of various metabolic processes, including electron transport in mitochondria and chloroplasts, photorespiration, and enzyme-catalyzed reactions. Superoxide radicals are involved in signaling pathways, including defense responses against pathogens and regulation of growth and development [109,110]. Hydrogen peroxide (H_2_O_2_) is a relatively stable ROS molecule that serves as a signaling molecule in plants. It is generated through the dismutation of superoxide radicals by superoxide dismutase or produced directly by enzymes such as NADPH oxidases (respiratory burst oxidase homologs, RBOHs). H_2_O_2_ is involved in various physiological processes, including stomatal closure, programmed cell death, seed germination, hormone signaling, and responses to environmental stresses such as drought, salinity, and pathogen attack. H_2_O_2_ also acts as a second messenger in signaling cascades involving mitogen-activated protein kinases (MAPKs) and transcription factors [110]. Hydroxyl radical (OH^•^) is a highly reactive and short-lived ROS molecule generated through the Fenton and Haber-Weiss reactions in the presence of transition metal ions and hydrogen peroxide. Due to its high reactivity, hydroxyl radicals can cause severe damage to cellular components. However, their direct production in plants is relatively limited. Instead, in the presence of ascorbic acid, the reaction of H_2_O_2_ with ferrous ions can indirectly generate hydroxyl radicals, leading to the formation of hydroxyl radicals in the apoplast. Hydroxyl radicals are associated with oxidative damage to proteins, lipids, and DNA [111]. Singlet oxygen (^1^O_2_) is a highly reactive form of oxygen that can be generated during photosynthesis under stress conditions, particularly in chloroplasts. It is produced when excited chlorophyll molecules transfer energy to oxygen molecules. Singlet oxygen is involved in photoinhibition of photosynthesis and oxidative damage to cellular components, including membrane lipids and proteins. However, in certain situations, singlet oxygen also acts as a signaling molecule regulating stress responses and developmental processes [112].

To mitigate the influence of reactive oxygen species (ROS), it is crucial to have an efficient way to reduce the damage caused by them. Therefore, plants have developed antioxidant defense mechanisms, such as superoxide dismutase (SOD), catalase (CAT), ascorbate peroxidase (APX), glutathione reductase (GR), peroxidase (POD), Phenylalanine ammonia-lyase (PAL), guaiacol peroxidase (GPOX), and plasma glutathione peroxidase (GSH-Px). These mechanisms help remove ROS and convert them into non-toxic substances, thereby maintaining cellular homeostasis [113]. Superoxide dismutase (SOD) is an essential antioxidant enzyme that catalyzes the dismutation of superoxide radicals (O^2•−^) into H_2_O_2_ and molecular oxygen (O_2_). SOD is present in various cellular compartments, including the cytosol, mitochondria, chloroplasts, and peroxisomes. It serves as the first line of defense against ROS by preventing the formation of highly reactive hydroxyl radicals [114]. Catalase (CAT) is an enzyme that contains heme and is primarily found in peroxisomes, but it is also present in other cell compartments such as the cytosol and mitochondria. CAT converts H_2_O_2_ into water (H_2_O) and molecular oxygen (O_2_). It plays a crucial role in detoxifying high levels of H_2_O_2_ generated during various metabolic processes, particularly in peroxisomes where high levels of ROS are produced [115]. Plant peroxidases are a diverse group of enzymes that play important roles in ROS detoxification and various physiological processes. They include ascorbate peroxidase (APX), peroxidase (POD), and glutathione peroxidase (GPX). Peroxidases utilize H_2_O_2_ as a substrate and reduce it while oxidizing various electron donors, such as ascorbate, glutathione, and phenolic compounds. These enzymes contribute to the removal of H_2_O_2_ and participate in cell wall lignification, hormone regulation, and defense responses [116,117]. Glutathione reductase (GR) is an enzyme that participates in the regeneration of reduced glutathione (GSH), which is an important antioxidant molecule. GR utilizes NADPH as a cofactor to reduce oxidized glutathione (GSSG) back to its reduced form (GSH). The ratio of GSH to GSSG is crucial for maintaining the cellular redox balance and is vital for various cellular processes, including ROS detoxification and modulation of cell signaling pathways [118]. Glutathione S-transferases (GSTs) are a diverse group of enzymes that catalyze the conjugation of glutathione (GSH) with various endogenous and exogenous compounds. GSTs play multifunctional roles in plants, including detoxification of exogenous substances, antioxidant stress protection, modulation of hormone signaling, and regulation of cellular redox status. They contribute to maintaining cellular redox homeostasis by participating in the recycling of oxidized glutathione and detoxification of electrophilic compounds [119].

Under abiotic stress, NAC (NAM, ATAF1/ATAF2, and CUC2) transcription factors can upregulate the expression of SOD and POD to help plants adapt to their environment [120,121]. Silencing of GRAS transcription factors significantly enhanced the expression of CAT, SOD, and POD [122]. MYB75, ZAT6, and other transcription factors regulate heavy metal stress by up-regulating GSH expression [123,124].

## 3. ROS in Humans

ROS can be generated from both endogenous and exogenous sources. Endogenous ROS production can be caused by immune cell activation, inflammation, ischemia, infection, cancer, excessive exercise, mental stress, and aging. Exogenous ROS, on the other hand, may be produced due to exposure to environmental pollutants, heavy metals (such as Cd, Hg, Pb, Fe, and As), certain drugs (such as cyclosporine, tacrolimus, gentamicin, and bleomycin), chemical solvents, cooking (such as bacon, used oils, and fats), cigarette smoke, alcohol, and radiation. When these exogenous compounds enter the body, they are degraded or metabolized, and ROS are produced as byproducts [125].

### 3.1. Cell Cycle and Development

The cell cycle is a cycle of stages that cells pass through to allow them to divide and produce new cells. It is a vital process in the development and growth of all organisms, including humans. ROS are highly reactive molecules that are produced as a by-product of normal cellular metabolism [126]. They play a crucial role in regulating the cell cycle and development. However, if ROS levels become too high, they may cause harm to cells (Figure 2B).

ROS are essential in the regulation of the cell cycle, particularly in the interphase stage. They act as signaling molecules that control the progression of cells through the cell cycle and help coordinate the processes of DNA replication and cell division. For instance, ROS can stimulate the activity of cell cycle protein-dependent kinases (CDKs). Additionally, many natural extracts such as acaciain, cordycepin, and bryophyllin can increase ROS levels in cancer cells, which inhibits the activity of CDK1, 2, 4, 6, B1, A, and E2-related cyclins, leading to S-phase or G2/M-phase arrest and apoptosis [127,128,129,130]. ROS can also stimulate the activity of transcription factors, which regulate gene expression and promote cell growth and division. Specifically, ROS can stimulate the activity of AP-1, a transcription factor that controls the expression of genes involved in cell growth, differentiation, and apoptosis (programmed cell death). Therefore, excessive ROS levels can promote cancer cell division by stimulating the activity of AP-1 [131,132,133]. NO also plays a role in regulating autophagy, where it acts as an inhibitory molecule. It can inhibit the generation of reactive oxygen species (ROS) through the mechanism of *S*-nitrosylation, leading to the prevention of autophagy [134]. ROS can also stimulate STAT3 activity, a transcription factor that regulates the expression of genes involved in cell growth, survival, and inflammation. However, an increase in ROS levels can inhibit STAT3 activity, leading to apoptosis in cancer cells [135,136,137]. ROS can also stimulate the activity of p53, a tumor suppressor protein that plays a crucial role in DNA repair and cell cycle regulation. An increase in ROS levels can activate p53, promoting cellular senescence and apoptosis [138,139,140]. 

Elevated ROS levels have been demonstrated to induce cell cycle arrest, senescence, and cancer cell death through the activation of the ASK1/JNK and ASK1/p38 signaling pathways in human fibroblasts and cancer cells. However, some evidence suggests that ROS-induced mutagenesis is a significant driver of tumorigenesis and progression [141,142,143]. Therefore, a dual role of ROS at the cellular level can be identified, and the effects of ROS on the cell cycle and development need to be viewed dialectically. Additionally, there is substantial evidence indicating that elevated ROS can inhibit mTOR/PI3K/AKT signaling, leading to apoptosis [144,145,146].

The coordinated regulation of the cell cycle is essential for proper cell growth and differentiation during development. In adults, ROS play a crucial role in the repair of damaged tissues and the maintenance of healthy cells. However, dysregulation of the cell cycle and ROS production can lead to the development of diseases such as cancer, in which cell division and growth cannot be controlled. In hypoxia, mitochondria release ROS that affect HIF1-α (hypoxia-inducible factor-1α) homeostasis, leading to increased transcription of angiogenic genes like vascular endothelial growth factor (VEGF). Furthermore, ROS promote ligand-independent H_2_O_2_-induced transactivation of VEGFR2 [147]. Collagen XVII (COL17A1) and transmembrane glycoprotein CD44 inhibit mitochondrial membrane potential and ROS production, promoting the formation of multilayered transformed epithelial cells [148]. The production of photocontrolled ROS activates the hair follicle stem cell niche, inducing cell proliferation and sustaining the growth phase over time [149].

### 3.2. Diseases

Excessive ROS production is associated with various diseases, such as cardiovascular disease, cancer, diabetes, neurodegenerative disease, and autoimmune disease [150]. ROS cause damage to DNA, proteins, and lipids in cells, which can lead to inflammation and tissue damage. The respiratory chain is not only central to energy production but is also a major source of intracellular ROS (Figure 2C). Many mitochondrial-localized proteins, including cytosolic adaptor protein p66shc, monoamine oxidases (MAOs), and NADPH oxidases 4 (NOX4), have been shown to contribute to mitochondrial ROS production [151]. Mitochondria-induced programmed cell death (PCD), or apoptosis, is a highly regulated process involving the release of pro-apoptotic proteins from mitochondria into the cytosol. This pathway plays a crucial role in maintaining tissue homeostasis, eliminating damaged cells, and is implicated in various diseases when dysregulated [152,153,154]. Most diseases caused by ROS are a result of abnormal mitochondrial function, such as impaired ATP production, increased reactive oxygen species (ROS) production, or disrupted mitochondrial membrane potential, which can disrupt the normal apoptotic signaling pathways. This can lead to altered cell survival and death decisions, potentially contributing to disease development [152,155]. Mitochondria are particularly abundant in cardiac tissue, and dysregulated ROS production and oxidative stress are associated with several heart diseases, including cardiac hypertrophy, heart failure (HF), ischemia-reperfusion injury (IRI), and diabetic cardiomyopathy [151]. ROS/RNS have been demonstrated to regulate multiple processes in skeletal muscle, such as transcription factor activity, ion transport, cell apoptosis, metabolism, and the physiological redox modulation of cysteine residues that participate in protein function regulation. For example, oxidation, nitrosylation, or *S*-glutathionylation of critical cysteine residues in the skeletal muscle RyR1 calcium release channel can significantly alter the channel’s mean open time and permeability [156]. In chronic obstructive pulmonary disease (COPD) caused by cigarette smoke, cigarette smoke extract (CSE) induces autophagy in mitochondria, leading to elevated ROS levels and the induction of cellular senescence in bronchial epithelial cells [157]. Interestingly, in the cardiac and pulmonary settings, miRNAs play a dual role in interfering with ROS and oxidative stress production. They do so directly by targeting oxidative and antioxidant enzymes (e.g., NOX, SOD, CAT), antioxidant genes (e.g., *Sirt*, *Trx1*), their transcription factors (e.g., Nrf2, FOXOs), and mitochondrial genes (e.g., *COX*, *Bnip3*). Indirectly, miRNAs target genes involved in apoptosis (e.g., *Bcl2*, *NF-κB*), tumor suppressor genes (e.g., *p53*), and interfere with pro-survival signaling pathways (e.g., Akt, IGF-1) [158].

Mitochondrial ROS can modify innate cellular responses both directly and indirectly. They directly alter cellular function by interacting with key transcription factors such as HIF-1α and nuclear factor κb (NF-κB). Indirectly, mitochondrial ROS enhance innate immune receptor sensing by interacting with other molecules such as mtDNA [159]. MFN2 is essential for the adaptation of mitochondrial respiration to stress conditions and the production of ROS. This production of ROS is associated with the induction of cytokines and nitric oxide, as well as dysfunctional autophagy, apoptosis, phagocytosis, and antigen processing [160]. Mitochondrial ROS contribute to T cell fate and function as signaling molecules. ROS synthesis stabilizes RORγt TF expression to promote Th17 differentiation. Targeting ROS production to eliminate auto-reactive T cells is a potential strategy to inhibit auto-reactive T cell activation without compromising systemic immune function [161]. Mitochondria contribute to ischemia-reperfusion (IR) injury by producing destructive ROS during reperfusion. This leads to the release of mitochondrial damage-associated molecular patterns (DAMPs), which activate the innate immune response and cause organ rejection [162]. Also, adoptive transfer of neutrophils from β-glucan-trained mice to naive recipients suppressed tumor growth in the latter in a ROS-dependent manner [163]. Mitochondria are considered one of the main targets of H_2_S signaling. It is known that H_2_S can directly impact the activity of the electron transport chain (ETC). Depending on its concentration, H_2_S can act as a substrate and a COX antagonist, protecting mitochondria from oxidative damage and enhancing their functionality. The interaction between H_2_S and ROS can influence mitochondrial homeostasis and overall cellular energy metabolism [164]. Chenodeoxycholic acid suppresses AML progression through promoting lipid peroxidation via the ROS/p38 MAPK/DGAT1 pathway and inhibiting M2 macrophage polarization [165]. ROS induces the separation of thioredoxin-interacting protein (TXNIP) from thioredoxin (TRX), which then binds and activates the NLRP3 inflammasome, resulting in subsequent cellular regulation and pyroptosis [166]. In addition, hypoxia-induced ROS has been found to stimulate NF-κB p65 phosphorylation, which is associated with the initiation of pyroptosis [167]. A novel regulator of Gasdermin D (GSDMD) mobilization that precedes pyroptosis is a xanthine oxidase ROS-activated MAP3K5/JNK2 substrate licensing complex [168]. ROS upregulates cysteine oxidative modification of GSDMD and promotes GSDMD cleavage [169]. Multiple studies have demonstrated that inhibition of the Xc-GSH-Gpx4 antioxidant system induces ferroptosis in cells [170].

### 3.3. System Signaling

Unlike in plants, ROS signals in humans typically act only in localized cells or organs and are not involved in systemic regulation. ROS acts as a second messenger for cell signaling and is essential for various biological processes in normal cells. In contrast to plants, humans require more timely and dynamic regulation, typically using neural and hormonal signals as the first messengers for signal transmission (Figure 2A). In the human body, electrical signals are transmitted at an average rate of 20–30 Hz to rapidly respond to various situations [15]. For example, the activation of splenic nerves can improve B cell responses and antibody production in response to protein immunization [171]. Hormones are transported through the bloodstream to regulate systemic effects, such as an increase in circulating glucocorticoids during stress, which can promote gluconeogenesis, mobilize amino acids, break down fats, and impair the body’s immune response [172]. This also results in a relatively lower tolerance to reactive oxygen species (ROS) in the human body, making them more sensitive to changes in ROS levels. RNS and ROS both serve as signaling molecules that regulate various cellular processes. The synergistic interaction between RNS and ROS can modulate signaling pathways involved in inflammation, immune response, cell apoptosis, and gene expression. For instance, the formation of peroxynitrite can modify proteins and activate specific signaling cascades [173].

One example of the relationship between systemic signaling and ROS is the role of ROS in inflammation. Inflammation is a key component of the immune response, involving the release of cytokines and the recruitment of immune cells to sites of injury or infection. ROS can act as signaling molecules in this process, helping to activate immune cells and stimulate cytokine production. ROS can also regulate the activity of other signaling molecules, such as growth factors and transcription factors, which are important regulators of cell growth and differentiation. For instance, ROS can activate transcription factors that stimulate the production of proteins involved in tissue repair and inflammation [174,175].

In the cardiovascular system, ROS have been proved to contribute to the development of hypertension, cause vascular hyperreactivity, endothelial dysfunction, vascular remodeling, influx of inflammatory cells into the vessel wall, increased wall stiffness, and fibrosis, which can be associated with blood pressure elevation. Additionally, ROS can stimulate the production of cytokines, which also contribute to the development of hypertension [176,177].

In the endocrine system, ROS have been shown to play a role in the regulation of insulin signaling, which is important for regulating blood glucose levels. ROS can interfere with the action of insulin, leading to the development of insulin resistance and type 2 diabetes [178,179]. 

In the skeletal system, ROS have been shown to play a role in bone metabolism, including the formation and maintenance of bone density. ROS can stimulate the production of cytokines that promote bone resorption or breakdown, as well as inhibit the production of proteins involved in bone formation [180,181]. IL-1β and FAC (iron ammonium citrate) both induce the generation of ROS and lead to the accumulation of lipid ROS and changes in the expression of iron death-related proteins in chondrocytes, thereby contributing to the progression of osteoarthritis [182].

Overall, the relationship between systemic signaling and ROS is complex, and further studies are needed to fully understand the mechanisms by which these molecules interact and regulate physiological processes in the body. Such studies may provide insights into disease development and the potential for therapeutic intervention.

### 3.4. ROS Hazard and Scavenging

ROS are highly reactive molecules that are byproducts of normal cellular metabolism and environmental stress. Similar to plants, humans generate reactive oxygen species (ROS) as natural byproducts of cellular metabolism, especially during processes such as mitochondrial respiration. ROS includes molecules such as superoxide radicals (O_2_^•−^), hydrogen peroxide (H_2_O_2_), and hydroxyl radicals (OH^•^). While ROS serve as signaling molecules with important roles, excessive ROS production can lead to oxidative stress, resulting in cellular damage, aging, and various diseases [125]. Mitochondria, known as the powerhouse of the cell, play a crucial role in cellular redox regulation. Mitochondrial respiration generates ROS as byproducts, particularly during the electron transport chain (ETC) involved in ATP production. The balance between ROS production and antioxidant defense systems is essential for maintaining mitochondrial redox homeostasis. Disruption of mitochondrial redox balance can lead to impaired mitochondrial function, compromised energy production, and increased oxidative stress [183]. The intracellular redox state serves as a critical signaling mechanism in human cells. ROS, particularly hydrogen peroxide (H_2_O_2_), function as signaling molecules to regulate cellular processes such as gene expression, cell growth, apoptosis (programmed cell death), and immune responses. ROS can modify specific cysteine residues in proteins, resulting in reversible oxidative modifications that impact protein structure and function. This redox signaling plays a pivotal role in cellular communication and adaptation to various physiological and environmental stimuli [184]. Imbalances in intracellular redox regulation can lead to oxidative stress, where the levels of reactive oxygen species (ROS) exceed the cellular antioxidant defense capacity. Oxidative stress is associated with the development and progression of numerous human diseases, including cardiovascular diseases, neurodegenerative diseases (such as Alzheimer’s disease and Parkinson’s disease), cancer, diabetes, and inflammation. Oxidative stress-induced damage to cellular components, such as DNA, proteins, and lipids, disrupts normal cellular functions and contributes to pathological conditions of diseases [183]. The body has multiple mechanisms to scavenge ROS and prevent oxidative stress, including enzymatic and non-enzymatic antioxidants (Figure 2B).

Similar to plants, different ROS molecules play distinct roles in various life processes. Superoxide radicals (O_2_^•−^) are primarily generated in mitochondria as byproducts of oxidative phosphorylation during cellular respiration. They can also be produced through various enzymatic reactions, including those involving NADPH oxidases and xanthine oxidases. Superoxide radicals play a role in cell signaling, including redox signaling pathways, immune responses, and gene expression regulation. However, their accumulation can lead to oxidative damage to proteins, lipids, and DNA [185]. Hydrogen peroxide (H_2_O_2_) is a relatively stable ROS molecule that can be formed by the dismutation of superoxide radicals through enzymatic reactions involving superoxide dismutase or by reactions involving oxidases and peroxidases. H_2_O_2_ acts as a signaling molecule in multiple cellular processes, including cell growth, differentiation, immune responses, and wound healing. It regulates redox-sensitive signaling pathways, such as those mediated by mitogen-activated protein kinases (MAPKs) and transcription factors. [186]. Hydroxyl radical (OH^•^) is the most reactive form of ROS, generated through the Fenton reaction or Haber–Weiss reaction. It causes significant damage to cellular components, including proteins, lipids, and DNA. The hydroxyl radical is involved in oxidative stress-related pathologies, including inflammation, aging, and various diseases. However, due to their short lifespan and high reactivity, their direct production in biological systems is limited [187]. Singlet oxygen (^1^O_2_) is generated through the interaction of the excited state of oxygen with various cellular components, such as photosensitizers, during processes like photosynthesis and photodynamic therapy. Singlet oxygen causes oxidative damage to lipids, proteins, and DNA. It is associated with phototoxicity and is utilized in targeted therapeutic approaches for the destruction of cancer cells [188]. Peroxynitrite (ONOO^−^) is formed by the reaction of superoxide radicals and nitric oxide (NO) under inflammatory and oxidative stress conditions. It is a highly reactive and cytotoxic molecule involved in the nitrosative stress pathway. Peroxynitrite can oxidize and damage proteins, lipids, and DNA, leading to various pathological conditions, including neurodegenerative diseases, cardiovascular diseases, and inflammation-related injuries [189].

Enzymatic antioxidants, such as superoxide dismutase and catalase, can directly neutralize ROS through chemical reactions. Non-enzymatic antioxidants, such as vitamin C and vitamin E, can also scavenge ROS, although they do so indirectly by providing electrons to ROS and being oxidized themselves in the process [190,191]. The two major sources of endogenous ROS are mitochondrial electron transport chains (ETCs) and NADPH oxidases (NOXs). The NOX family includes NOX1, NOX2, NOX3, NOX4, NOX5, and dual oxidases (DUOX1 and DUOX2) [192]. The main intracellular enzymatic antioxidants are superoxide dismutase (SOD) (cytosolic SOD1, mitochondrial SOD2, and extracellular SOD3), peroxidase (PRX), glutathione peroxidase (GPX), and catalase (CAT) [193]. Superoxide dismutase (SOD) is an important antioxidant enzyme that catalyzes the dismutation of superoxide radicals (O_2_^•−^) into hydrogen peroxide (H_2_O_2_) and molecular oxygen (O_2_). There are three isoforms of SOD: copper-zinc SOD (SOD1 or Cu/Zn-SOD), manganese SOD (SOD2 or Mn-SOD), and extracellular SOD (SOD3 or EC-SOD). SOD is located in different cellular compartments and helps prevent the accumulation of superoxide radicals, which can lead to oxidative stress [194]. Catalase is an enzyme that is primarily present in peroxisomes, but it is also found in other cellular compartments such as the cytosol and mitochondria. Catalase plays a crucial role in the decomposition of H_2_O_2_ into water (H_2_O) and molecular oxygen (O_2_). By rapidly breaking down H_2_O_2_, catalase helps protect cells from oxidative damage and prevents the formation of highly reactive hydroxyl radicals [195]. Glutathione peroxidase (GPx) is a group of enzymes that utilize the reducing power of glutathione (GSH) to convert H_2_O_2_ and organic hydroperoxides into water (H_2_O) and corresponding alcohols. GPx enzymes are selenium-dependent and exist in various isoforms, including cytosolic GPx (GPx1), gastrointestinal GPx (GPx2), and phospholipid hydroperoxide GPx (GPx4), among others. GPx enzymes play a critical role in protecting cells and tissues from oxidative damage [196]. Glutathione reductase (GR) is an essential enzyme involved in the regeneration of reduced glutathione (GSH), which is a major cellular antioxidant. GR utilizes NADPH as a cofactor to convert oxidized glutathione (GSSG) back into its reduced form (GSH), ensuring an ample supply of GSH for various redox reactions. GR is crucial for maintaining the oxidative–reductive balance and the antioxidant defense system within cells [197]. Thioredoxin reductase (TrxR) is an enzyme that participates in the reduction of oxidized proteins through the thioredoxin system. They utilize NADPH to regenerate the reduced form of thioredoxin (Trx), which serves as an oxidoreductase involved in various cellular processes, including DNA synthesis, protein folding, and redox signaling. TrxR enzymes play a critical role in maintaining proteins in a reduced state, protecting them from oxidative modifications [198].

ROS formation is subtly regulated by DNA repair enzymes and antioxidant defense systems within the human body, which can regenerate damaged antioxidant molecules. Nuclear factor E2-related factor 2 (NRF2), an intracellular transcription factor that controls the expression of antioxidant genes and protects cells from oxidative and electrophilic stress, is oxidized when intracellular ROS accumulates abnormally, thereby blocking its interaction with NRF2 and subsequent degradation. The stable NRF2 protein is then transferred to the nucleus, where it binds to antioxidant response elements (ARE) and activates the transcription of enzymatic antioxidants (such as CAT, PRX, and GPX) and enzymes involved in GSH metabolism [199]. Some tumor suppressor genes (*BRCA1*, *BRCA2*, *TP53*, *PTEN*, *FXOP3,* and *ATM*) are also involved in ROS scavenging [200].

In addition to the natural antioxidant defense system within the human body, there are many exogenous antioxidants that can help scavenge ROS and prevent oxidative stress. These include dietary sources, such as fruits and vegetables, as well as supplements, such as vitamins and minerals. There is some evidence that proves that a diet rich in antioxidants may be protective against certain diseases, such as cancer, heart disease, and neurodegenerative diseases [201,202,203,204]. However, it is important to note that more research is needed to fully understand the role of antioxidants in disease prevention. Additionally, it is not always clear whether the benefits of antioxidant supplements are the same as those of a diet rich in antioxidant foods. 

Overall, the scavenging of ROS is an essential process in the body that helps prevent oxidative stress and maintain cellular health.

## 4. The Crosslink of ROS in Plants and Humans

Both plants and humans experience ROS-based oxidative stress events, and oxidation is the most direct and straightforward result when external conditions change. Consequently, both plants and humans have evolved regulatory mechanisms that utilize ROS as a signal and ROS scavenging systems (Figure 3). However, the impact of ROS and their scavenging mechanisms differ significantly between plants and humans.

### 4.1. Diverse ROS and Their Functions

Plants possess an additional source of ROS production through photosynthesis, where excess energy can lead to the generation of O_2_^•−^ and H_2_O_2_, particularly in chloroplasts [205]. This unique process sets plants apart from humans, who do not undergo photosynthesis. Instead, human ROS production is primarily associated with mitochondrial respiration, leading to the formation of O_2_^•−^ [206]. Due to the presence of photosynthesis, plants have developed stronger ROS tolerance and clearance mechanisms compared to humans. At the same time, there are two types of NADPH oxidases (NOX enzymes) and peroxisomes that can generate ROS primarily consisting of O_2_^•−^ and H_2_O_2_, playing roles in cell signaling and various metabolic processes [126,207,208,209].

Different types of ROS typically interact in complex ways, with their effects depending on concentration, location, and the overall cellular environment. Moreover, various ROS types have distinct functions. In plants, H_2_O_2_ acts as a second messenger in various signaling pathways related to growth, development, and stress responses, while also regulating stress mitigation against non-biological stressors [210]. O_2_^•−^ is produced as a defense response against pathogens and can initiate allergic reactions to limit pathogen spread [211]. In humans, H_2_O_2_ serves as a redox signaling molecule, participating in the regulation of cellular processes, including proliferation and differentiation [212]. O_2_^•−^ is produced during immune cell respiration bursts to eliminate pathogens [213], while OH^•^ is associated with DNA damage and aging-related processes [214]. 

However, due to current experimental conditions and limitations of experimental materials, studying highly reactive ROS molecules like OH^•^ in the cellular microenvironment remains challenging. Additionally, there are dynamic interconversions among ROS molecules, and coordinated factors play a role in their functional effects, leading some studies to approach ROS as a collective entity. In the future, conducting more specific research on the functions of different ROS will help us better understand the role of ROS in maintaining cellular health and function.

### 4.2. Antioxidant Enzymes

Many enzymes encoded by the ROS network function in defined ROS removal pathways, such as the SOD-Asada-Foyer-Halliwell pathway (also known as the SOD-ascorbic acid-glutathione pathway) found in chloroplasts, mitochondria, cytoplasmic lysates, and peroxisomes. The fact that SOD has evolved independently in at least three different contexts, including Fe-/Mn-SOD, CuZn-SOD, and Ni-SOD, highlights the important role played by SOD enzyme activity in different biological systems. Additionally, the discovery of Fe-, Mn-, and CuZn-SODs in mosses, spruce, and higher plants further reinforces the importance of SOD in the protection of photosynthetic multicellular organisms [215].

Research data indicate that the levels of antioxidant enzymes in plants could be 5–10 times higher than those in humans. Furthermore, under stress treatment, the activity of plant antioxidant enzymes could increase by 2–5 times compared to their normal levels [216,217,218,219]. Indeed, compared to terrestrial plants, terrestrial animals possess a unique class of selenoproteins that enhance their ROS clearance capacity through the incorporation of selenocysteine into their catalytic centers. Examples of such selenoproteins include thioredoxin reductases (TR) and glutathione peroxidases (GPx). These selenoproteins play vital roles in maintaining redox balance and protecting cells from oxidative damage in animals by efficiently scavenging ROS. The presence of these specialized selenoproteins contributes to the enhanced ROS clearance capability observed in terrestrial animals [220].

### 4.3. Intracellular ROS Production 

In both plants and animals, ROS are generated within various cellular compartments, including the mitochondria, peroxisomes, cytosol, and chloroplasts (in plants). The intracellular distribution of ROS can vary depending on the specific cellular processes and environmental conditions. Due to limitations in current experimental methods, measuring the levels of reactive oxygen species (ROS) in various organelles poses certain challenges. However, in vitro experiments and estimation based on chemical potentials still allow for the inference of ROS concentrations in certain organelles.

In the normal state of mitochondria, the actual reduction potential (Eh) for the reduction of O_2_ to O_2_^•−^ varies with the relative concentrations of O_2_ and O_2_^•−^. The concentration of O_2_ within the mitochondria ranges from 3 to 30 μM, and due to the presence of MnSOD in the mitochondrial matrix, O_2_^•−^ is dismutated to H_2_O_2_, resulting in relatively low levels of O_2_^•−^, estimated to be within the range of 10–200 pM [221]. Classic experiments with mitochondrial electron transport chain (mETC) inhibitors indicate that Complex I and Complex III are the main sources of mitochondrial and cellular ROS. Specifically, electrons can undergo reverse electron transfer (RET), reducing NAD^+^ to NADH and generating O_2_^•−^, as well as the accumulation of electrons inside CIII, which reach the CIII_o_ site, producing O_2_^•−^ [222]. There are two modes of O_2_^•−^ generation in mitochondria: one is when mitochondria do not produce ATP, resulting in a high Δp (proton motive force) and reduced CoQ (coenzyme Q) pool, and the other is when the NADH/NAD^+^ ratio is high in the mitochondrial matrix. In mitochondria actively producing ATP, the amount of O_2_^•−^ generated is much lower than in the first two modes [221].

In the chloroplasts specific to plants, PSII can generate superoxide anions (O_2_^•−^), hydrogen peroxide (H_2_O_2_), and hydroxyl radicals (OH^•^). Electron leakage on the electron acceptor side of PSII leads to the production of O_2_^•−^, which is then dismutated to H_2_O_2_. H_2_O_2_ is further converted to OH^•^ by non-heme iron. On the electron donor side of PSII, incomplete water oxidation results in the formation of H_2_O_2_, which is further reduced to OH^•^. In PSI, the stepwise production of O_2_^•−^ and H_2_O_2_ depends on ineffective photochemical and non-photochemical quenching (NPQ) [223].The steady-state concentration of O_2_ inside intact chloroplasts in the light depends on the external concentration of O_2_. At low external O_2_ concentrations (30 µM), the ratio of the internal to the external is about five, whereas at concentrations corresponding to those in air-saturated water (≈258 µM), the O_2_ concentration of isolated chloroplasts is similar to that of the medium. The O_2_ consumption associated with O_2_ reduction by PSII membranes capable of water-splitting is about 1 µmol O_2_ (mg Chl)^−1^ h^−1^ or 4 µmol O_2_^•−^ (mg Chl)^−1^ h^−1^ when O_2_ is the only electron acceptor. Highly lipophilic peroxides (LOOH) and relatively hydrophilic ones (ROOH) were distinguished by the rate of reaction with a specific fluorescence probe (Spy-HP). The formation rates of both LOOH and ROOH were estimated to be 0.022 µmol LOOH (µmol RC)^−1^ s^−1^ and 1.11 µmol ROOH (µmol RC)^−1^ s^−1^ [224].

In plant peroxisomes, H_2_O_2_ can be generated through various peroxisomal metabolic processes, including photorespiration, β-oxidation, superoxide dismutation, sulfite oxidation, polyamine degradation, and others. Peroxisomes are equipped with an efficient set of H_2_O_2_ scavengers, including catalase (CAT) in the matrix and ascorbate peroxidase (APX) on the membrane [225]. Among them, the rate of H_2_O_2_ production by peroxisomal glycolate oxidase (GOX) is reported to be approximately 2 times higher in chloroplasts and 50 times higher in mitochondria [226]. In mammals, peroxisomes play a crucial role in cellular H_2_O_2_ metabolism and contain various FAD-containing oxidoreductases that can generate H_2_O_2_ as part of their normal catalytic activity. They may also contain xanthine oxidoreductase (XDH) and inducible nitric oxide synthase (NOS2), producing O_2_^•−^ and NO^•^, respectively [227]. Peroxisomes in rat liver may account for up to 20% of oxygen consumption and 35% of H_2_O_2_ production [228]. Furthermore, peroxisomes in different organs perform distinct functions, and we recommend referring to the study by Islinger [229] for more information on this topic.

While mitochondria, chloroplasts (in plants) and peroxisomes are major contributors to ROS production in cells, other regions such as the cytosol [230], lysosomes [231], and endoplasmic reticulum (ER) [232] can also generate ROS to a lesser extent. It is important to note that the distribution and levels of ROS can vary depending on cell type, tissue, and physiological conditions in both plants and animals. The regulation of ROS generation and scavenging systems is crucial for maintaining cellular redox balance and preventing oxidative stress-induced damage in these organisms.

### 4.4. Application of Exogenous Hydrogen Peroxide

As mentioned earlier, ROS can act as signaling molecules in cellular homeostasis. Among them, H_2_O_2_ is a relatively stable ROS molecule that plays a role as a signaling molecule in both plants and animals. It can be rapidly cleared by catalase (CAT), making it the preferred substance for researchers to externally apply ROS molecules. In the process of cell proliferation, a certain level of H_2_O_2_ can promote cell growth, depending on the baseline concentration of the cells. Additionally, it also regulates downstream MAPKs, demonstrating the versatility of ROS in cellular functions.

Researchers commonly utilize varying concentrations of H_2_O_2_ to examine its impact. In studies involving single-cell algae, plant tissue cultures, and seed treatments, a preference is observed for low concentrations of H_2_O_2_, typically not exceeding 10 mM [8,233,234]. When treating whole plants, the application of H_2_O_2_ is more aggressive, ranging from several millimolar (mM) concentrations to several hundred mM [235,236]. This may be related to the ease of H_2_O_2_ entry into cells. In *Chlamydomonas*, compared to the control group, a significant impact on the cell cycle can be observed with a concentration of H_2_O_2_ that is only half of the normal concentration [8]. Therefore, applying exogenous H_2_O_2_ at different concentrations of ROS substrates during different stages of plant cell development is a feasible method. 

In animal experiments, the concentrations of exogenously applied H_2_O_2_ are generally relatively low. They are typically applied to the wound site and range from several micromolars (µM) to several millimolars (mM). These concentrations have shown the ability to promote wound healing [237,238,239]. By comparison, we can observe that animals have relatively lower tolerance to reactive oxygen species (ROS) compared to plants. This difference may be attributed to the fact that plants generate ROS during photosynthesis, which contributes to their higher tolerance towards ROS [104].

### 4.5. ROS Signal

Plants have a unique cascade amplification mechanism that allows the amplification of ROS signals through RBOH, transmitting them throughout the entire plant. This is an important feature evolved by plants within their limited signal transmission modes, enabling them to respond to changes in the external environment relatively quickly. The majority of studies conducted in animal cells address RIRR (ROS-induced ROS release) pathways and mechanisms used to communicate between different cellular compartments within the cell. In contrast, very little is known in animal cells about RIRR communication between cells [208]. According to the existing literature, it is known that ROS can participate in the signaling regulation of local tissues. For instance, ROS can modulate the tumor microenvironment, thereby accelerating tumor proliferation. [240,241,242]. It is possible that in animals, there is a lack of cascading amplification mechanisms for ROS, which leads to the inability of ROS generated by tissues to be suitable for long-range signal transmission due to their highly reactive chemical nature, thus, some secondary products of ROS mediate inflammatory signaling in a variety of cellular environments. For example, lipid peroxidation products, such as 4-hydroxynonenal, and oxidized phospholipids (OxPLs) regulate NF-κB activation and inflammatory signaling through numerous pathways [243].

### 4.6. Diverse Consequences of High ROS Levels

High levels of ROS can have different consequences in plants and humans. In plants, elevated ROS levels can lead to oxidative damage, mitochondrial- and chloroplast-mediated programmed cell death (PCD), and mitochondrial-mediated necrotic cell death and impaired growth [58,59,244]. In humans, excessive ROS production can contribute to the development of various diseases, including cardiovascular diseases, neurodegenerative diseases, cancer, and age-related disorders [243,245,246]. At the cellular level, in addition to plant-like mitochondria-mediated programmed cell death (PCD), it also includes pyroptosis, parthanatos, and ferroptosis induced by excessive ROS [153,154,170].

In summary, while both plants and humans generate ROS as byproducts of cellular metabolism, there are differences in their production mechanisms, defense systems, functional roles, and impacts on the organism. Understanding these similarities and differences is important for studying ROS-related processes and developing strategies to manage oxidative stress in both plants and humans.

## 5. Conclusions and Perspective

Life-forms present on Earth today have evolved from ancient common ancestors in the presence of ROS for at least 3.6–3.8 billion years, underscoring the close relationship between many physiological/developmental processes and ROS today. Furthermore, many significant events in the evolution of life on Earth, such as the emergence of eukaryotes and the endosymbiotic events leading to the formation of chloroplasts and mitochondria, may have taken place in the presence of ROS [215]. 

Over the years, the identification of commonalities between plants and humans has remained a significant challenge. However, several important discoveries have shed light on the interconnectedness of certain elements, particularly those related to reactive oxygen species (ROS). For instance, the porphyrin skeleton found in chlorophyll and protoheme (a group of photosensitizers based on porphyrin) has been found to generate ROS, making it essential for the photodynamic therapy of human diseases. Additionally, both plants and humans share glutathione, a crucial component of the intracellular antioxidant system, as well as thioredoxin and peroxidase. In recent studies, melatonin, known for its antioxidant and anti-inflammatory properties in humans [247,248], has emerged as a novel plant hormone capable of regulating the activities of antioxidant enzymes [249]. This discovery further emphasizes the interconnectedness between plants and humans.

The future development of ROS research can be described in four aspects as below. 

Natural antioxidants derived from plants have shown promising potential in targeting various human diseases. In addition to well-known natural metabolites such as vitamin C [250], tea polyphenols [251], curcumin [252], anthocyanidin [253], and coenzyme Q10 [254], they have been proven to combat several human diseases. Bioactive peptides obtained from plants through enzymolysis have also emerged as valuable compounds for use in food, cosmetics, and pharmaceuticals due to their antioxidative properties. For example, lunasin peptide has been identified as an effective bioactive peptide with antioxidant properties [255]. Utilizing bioactive peptides derived from agricultural residue can be a favorable choice as by-products. Recent research has shown that the plant protein RDR1 exhibits potential in addressing human cancer [256]. To recognize the numerous genes and proteins participating in ROS-related pathways such as MAPK, CDKs, mTOR and GSH in plants can be researched to fight human cancer.The introduction of mammalian genes into plants has shown promise in enhancing stress resistance and increasing crop yield. For instance, the expression of a detoxifying transgene, mammalian cytochrome P450 2e1, in the houseplant Epipremnum aureum (pothos ivy) has resulted in genetically modified plants with effective detoxifying activity against benzene and chloroform [257]. In addition, the transgenic expression of FTO in rice and potato has demonstrated significant yield and biomass increases of approximately 50%. [258]. These findings suggest that in the future, the expression of human reactive oxygen species-related genes in plants could potentially enhance their ability to resist oxidative stress and improve crop yield.Apart from reactive oxygen species (ROS), there are other reactive compounds found in both plants and humans that have been shown to exert influence on stress responses and human health. These include reactive nitrogen species (RNS), reactive sulfur species (RSS), and molecular hydrogen (H_2_). ROS and RNS have been extensively studied and are known to play roles in various pathological processes, including Alzheimer’s disease, cancer, diabetes mellitus, cardiovascular disease, and aging [259]. The reactive sulfur species (RSS) are crucial for maintaining redox homeostasis and act as antioxidants and free-radical scavengers, contributing to overall cellular health and balance [260,261,262,263]. In humans, molecular hydrogen functions as a therapeutic antioxidant by selectively reducing hydroxyl radicals [264,265,266]. In plants, hydrogen plays a crucial role in plant hormone signal transduction and stress response mechanisms. [267,268]. While the influence of these reactive compounds on plants and humans is well-established, not all of the underlying mechanisms linking these signals have been thoroughly investigated.Research targeting reactive oxygen species (ROS) in plants and humans shares common areas of investigation. The following aspects are noteworthy: (a) Hypoxia: Hypoxia is known to be detrimental in cancer and contributes to its progression. Post-hypoxic tumor cells exhibit an ROS-resistant phenotype that enhances their survival in the bloodstream and facilitates the establishment of overt metastasis [269]. However, compared to animal studies, there is a relative dearth of research on hypoxia in plants [270,271,272,273,274]. Further investigation is needed to understand the molecular mechanisms underlying how plants adapt to and tolerate hypoxic stress caused by flooding, as well as how they sense oxygen signals and mitigate hypoxia-induced stress. (b) Ion involvement: Various ions, such as Fe^2+^, Mg^2+^, Ca^2+^, Se^4+^, Zn^2+^, and Cu^2+^, participate in enzyme catalysis. For example, highly reactive radical OH^•^ is generated from H_2_O_2_ through the Fenton or Weise reaction in the presence of catalytically active metals like Fe^2+^ and Cu^+^ [275,276]. However, the detailed redox mechanisms of electrical charge transfer in both humans and plants are not yet fully understood. (c) Nanomaterial development: Nanomaterials based on ROS have been developed to serve as antioxidants in plants and humans. A CuSe/CoSe_2_ nanoflower-like heterostructure has been synthesized, which can consume excess GSH in the tumor microenvironment (TME) and decompose H_2_O_2_ to produce ·OH, leading to effective ferroptosis [277,278]. Cerium oxide nanoparticles (CeO_2_ NPs) exhibit superoxide dismutase (SOD)-like activities, enabling them to catalyze the decomposition of ROS [279]. These findings highlight the promising potential of utilizing ROS-scavenging nanozymes to enhance the inherent antioxidant functions in both humans and plants.

Understanding the intricate relationship between plants and humans in the context of ROS opens up exciting possibilities for improving human health, agricultural practices, and environmental sustainability.

## Figures and Tables

**Figure 1 ijms-24-13052-f001:**
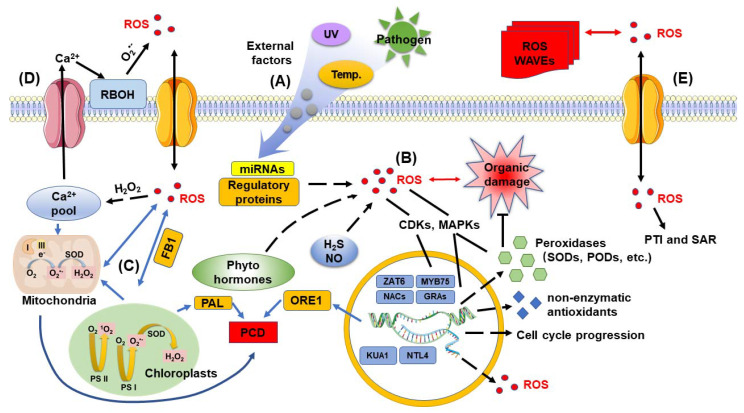
External factors generate ROS directly/indirectly (e.g., heavy metals, UV) in plants, which then regulate functions via CDKs/MAPKs affecting transcription and the cell cycle, and control cell death via calcium, mitochondria, and chloroplasts, with ROS waves activating defense. (**A**) External environmental factors can generate ROS directly or indirectly, such as heavy metal ions, ultraviolet radiation, etc., which generate ROS through oxidation. Some pathogens can generate ROS by stimulating miRNAs, regulatory proteins, and phytohormones. (**B**,**C**) After ROS is produced, it mainly regulates plant-related functions through two pathways. The first pathway (**B**), as indicated by the black line, activates or inhibits the activity of CDK- and MAPK-related proteins due to the concentration of ROS, thus affecting the activity of downstream transcription factors and the expression of related genes, and ultimately controlling peroxidases, non-enzymatic antioxidants, and related events in the cell cycle progression. The second pathway (**C**), as indicated by the blue line, controls programmed cell death by affecting calcium pools, mitochondria, and chloroplasts. (**D**) Calcium ions are transported through calcium channels to activate RBOH and produce ROS in the extracellular space. ROS can freely enter and exit cells through water channel proteins. (**E**) ROS waves can be formed to transmit throughout the plant, activating PTI and SAR. “Parts of the figure were drawn by using pictures from Servier Medical Art. Servier Medical Art by Servier is licensed under a Creative Commons Attribution 3.0 Unported License (https://creativecommons.org/licenses/by/3.0/ (accessed on 20 October 2022))”.

**Figure 2 ijms-24-13052-f002:**
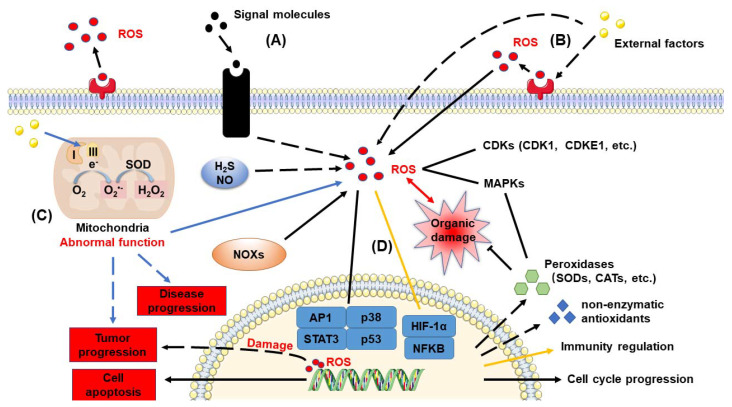
ROS act as messengers to affect regulating pathways and control immunity, triggered by external factors. Meanwhile, mitochondrial issues cause ROS leakage, leading to diseases and apoptosis. (**A**) As indicated by the black line, certain signaling molecules promote ROS production via receptor activation of associated proteins, and subsequently control the function of relevant pathways by affecting downstream CDKs, MAPKs, and transcription factors. (**B**) Some external factors can also lead to ROS production. (**C**) As shown by the blue line, mitochondrial dysfunction can cause ROS leakage, resulting in the occurrence of diseases and tumors, and ultimately leading to cell apoptosis. (**D**) As indicated by the orange line, ROS can regulate immune function by controlling transcription factors. “Parts of the figure were drawn by using pictures from Servier Medical Art. Servier Medical Art by Servier is licensed under a Creative Commons Attribution 3.0 Unported License (https://creativecommons.org/licenses/by/3.0/ (accessed on 20 October 2022))”.

**Figure 3 ijms-24-13052-f003:**
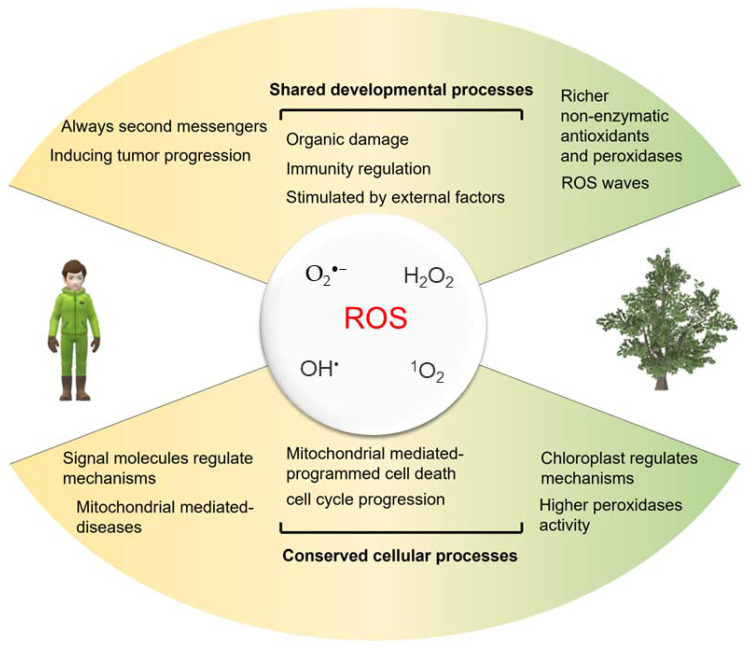
The similarities and differences in ROS functions between plant and human. The yellow and green sections on the left and right represent the unique ROS functions in plants and humans, respectively. The middle section represents the shared ROS functions between the two.

## Data Availability

No new data were created or analyzed in this study. Data sharing is not applicable to this article.

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
