# Peer review of "Reactive Oxygen Species: A Crosslink between Plant and Human Eukaryotic Cell Systems"

_ijms, 2023, doi:10.3390/ijms241713052_

Round 1

Reviewer 1 Report

1. Manuscript title: I don't understand what you mean by "a hinge". In my opinion, the manuscript title should point out the main theme and should be based on well-recognized findings. The suggestion is to include the part of share function and conserved mechanisms of ROS between plants and humans.

2. Abstract: I must say that this part is too weak and it should be improved by adding some key thoughts to this review article and future directions. The authors should add several sentences to summarize the universal machinery of ROS.

3. Keywords: These should be key terms but did not appear in the manuscript title.

4. L16-17: I don't understand what you mean by "have been involved".

5. L17: "It has been proved that ROS play critical roles in both plants and humans." Please specify the roles.

6. L731: What do you mean by "contrast"?

7. Conclusion and perspective: Too long, and needs to be concentrated, only needs to give a summary of the most important points.

8. L991: It's very strange to use "in conclusion" in "conclusion and perspective".

9. There are several sources of ROS, so the authors should clarify their roles and mechanisms in plants and humans.

10. Why do the authors attempt to compare plants with humans but not with animals?

Author Response

Dear reviewer:

Thank you for your comments on our manuscript entitled “Reactive oxygen species: A hinge between plants and humans” (Manuscript ID:  ijms-2555884). Those comments were very helpful for improving our manuscript. We believed we have answered reviewer’ s all comments. Based on your recommendations, we have made further modifications to the article, particularly in the title, “abstract” and “conclusion and perspective”. We have enriched the depiction of the various types of reactive oxygen species in both plants and humans. Below the comments of the reviewers are responded point by point and the revisions article are indicated by red.

Reviewer 1

Comment 1: Manuscript title: I don't understand what you mean by "a hinge". In my opinion, the manuscript title should point out the main theme and should be based on well-recognized findings. The suggestion is to include the part of share function and conserved mechanisms of ROS between plants and humans.

Reply 1: Thank you for your advice. We understand that the title is not sufficient to fully summarize the content of the entire review, taking into account the comments of yours and reviewer 2, we have changed the title to “Reactive oxygen species: A crosslink between plants and humans eukaryotic cells systems” to improve the quality of title.

Comment 2: Abstract: I must say that this part is too weak and it should be improved by adding some key thoughts to this review article and future directions. The authors should add several sentences to summarize the universal machinery of ROS.

Reply 2: Thank you for your suggestions. We rewrite the abstract as follows.

Reactive oxygen species (ROS) are important regulating factors that play a dual role in plant and human cells. As the first messenger response in organisms, ROS coordinate signals in growth, development, and metabolic activity pathways. They also can act as an alarm mechanism, triggering cellular responses to harmful stimuli. However, excess ROS cause oxidative stress-related damage, oxidize organic substance, leading to cellular malfunctions. This review summarizes the current research status and mechanisms of ROS in plant and human eukaryotic cells, highlighting the differences and similarities between the two and elucidating their interactions with other reactive substances and ROS. Based on the similar regulatory and metabolic ROS pathways in the two kingdoms, this review proposes future developments that can provide opportunities to develop novel strategies for treating human diseases or creating greater agricultural value.

Comment 3: Keywords: These should be key terms but did not appear in the manuscript title.

Reply 3: Thank you for reminding us. We realized that the key words are not well fitted with the title. We have changed the key terms to “reactive oxygen species (ROS), regulatory factors, oxidative stress, cellular response, ROS scavenging”.

Comment 4: L16-17: I don't understand what you mean by "have been involved".

Reply 4: Thank you for your reminder. We want to express that ROS participate in various life activities, while “have been involved” may makes the role of ROS misunderstand. We have rewritten the abstract and tried to avoid such misunderstanding.

Comment 5: L17: "It has been proved that ROS play critical roles in both plants and humans." Please specify the roles.

Reply 5: Thank you for your suggestion. We have enriched the sentences as follows: As the first messenger response in organisms, ROS coordinate signals in growth, development, and metabolic activity pathways. They also can act as an alarm mechanism, triggering cellular responses to harmful stimuli. However, excess ROS cause oxidative stress-related damage, oxidize organic substance, leading to cellular malfunctions.

Comment 6: L731: What do you mean by "contrast"?

Reply 6: Thank you for your reminder, we realized that this word “contrast” is not appropriate in this position. We aim to summarize the similarities and differences of ROS (Reactive Oxygen Species) in both plant and human contexts. Following your suggestions, we have replaced the term "contrast" with "crosslink" to better align with the content of the article and enhance reader comprehension.

Comment 7: Conclusion and perspective: Too long, and needs to be concentrated, only needs to give a summary of the most important points.

Reply 7: Thank you for your advice. We have deleted a lot of sentences and examples to highlight our points. The paragraph 901-908 has been deleted. We also delete some sentences, like 919-921, 930-933, 945-946, 948-953, 955-956, 970-971, 973-977, 982-986, 988-989, 991-1003.

Comment 8: L991: It's very strange to use "in conclusion" in "conclusion and perspective".

Reply 8: Thank you for your kind reminder. We changed the contents of “conclusion and perspective” and tried to avoid this kind of mistake.

Comment 9: There are several sources of ROS, so the authors should clarify their roles and mechanisms in plants and humans.

Reply 9: Thank you for the insightful advice. Indeed, distinct types of reactive oxygen species (ROS) play varying roles, each contributing unique functions within biological systems. While our text strives to encompass the functions of specific ROS variants as outlined in the referenced materials, it is noteworthy that in some instances within the literature, researchers adopt a broader perspective by collectively addressing ROS functions. This approach might stem from current limitations in research methodologies during this stage.

Considering this, we have revisited and revised section '4.1 Diverse ROS and their functions' spanning from L754 to L780. Our intention is to provide readers with an enriched understanding of the diverse roles carried out by different ROS species. Furthermore, we have taken the opportunity to briefly introduce the existing challenges encountered in the study of ROS. By doing so, we aspire to enhance the discourse surrounding the functions of distinct ROS varieties in both plants and animal contexts.

The rewrite text is under below:

4.1 Diverse ROS and their functions

Plants possess an additional source of ROS production through photosynthesis, where excess energy can lead to the generation of ^1O2 and H2O2, particularly in chloroplasts [1]. This unique process sets plants apart from humans, who do not undergo photosynthesis. Instead, human ROS production is primarily associated with mitochondrial respiration, leading to the formation of O2•− [2]. Due to the presence of photosynthesis, plants have developed stronger ROS tolerance and clearance mechanisms compared to humans. At the same time, there are two types of NADPH oxidases (NOX enzymes) and peroxisomes that can generate ROS primarily consisting of O2•− and H2O2, playing roles in cell signaling and various metabolic processes [3-6].

Different types of ROS typically interact in complex ways, with their effects depending on concentration, location, and the overall cellular environment. Moreover, various ROS types have distinct functions. In plants, H2O2 acts as a second messenger in various signaling pathways related to growth, development, and stress responses, while also regulating stress mitigation against non-biological stressors [7]. O2•− is produced as a defense response against pathogens and can initiate allergic reactions to limit pathogen spread [8]. In human, H2O2 serves as a redox signaling molecule, participating in the regulation of cellular processes, including proliferation and differentiation [9]. O2•− is produced during immune cell respiration bursts to eliminate pathogens [10], while OH is associated with DNA damage and aging-related processes [11].

However, due to current experimental conditions and limitations of experimental materials, studying highly reactive ROS molecules like OH in the cellular microenvironment remains challenging. Additionally, there are dynamic interconversions among ROS molecules, and coordinated factors play a role in their functional effects, leading some studies to approach ROS as a collective entity. In the future, conducting more specific research on the functions of different ROS will help us better understand the role of ROS in maintaining cellular health and function.

Comment 10: Why do the authors attempt to compare plants with humans but not with animals?

Reply 10: We appreciate your insightful comments on our manuscript. As one of the most highly evolved mammals on land, humans possess complex regulatory mechanisms. Additionally, many animal disease models are based on humans, making the study of ROS-induced diseases particularly extensive and thorough. For instance, many human cancers are characterized by uncontrolled ROS regulation, which leads to tumor progression. We propose that understanding the ROS regulatory mechanisms in certain plants could help control tumor-associated ROS and inhibit tumor progression.

Our aim is to reveal new perspectives based on the similarities and differences in ROS regulation between plants and humans, with the goal of generating novel benefits.

  1. Riaz, A.; Deng, F.; Chen, G.; Jiang, W.; Zheng, Q.; Riaz, B.; Mak, M.; Zeng, F.; Chen, Z.H. Molecular Regulation and Evolution of Redox Homeostasis in Photosynthetic Machinery. Antioxidants (Basel) 2022, 11, doi:10.3390/antiox11112085.
  2. Musicco, C.; Signorile, A.; Pesce, V.; Loguercio Polosa, P.; Cormio, A. Mitochondria Deregulations in Cancer Offer Several Potential Targets of Therapeutic Interventions. Int. J. Mol. Sci. 2023, 24, doi:10.3390/ijms241310420.
  3. Kleiboeker, B.; Lodhi, I.J. Peroxisomal regulation of energy homeostasis: Effect on obesity and related metabolic disorders. Mol. Metab. 2022, 65, 101577, doi:10.1016/j.molmet.2022.101577.
  4. Sandalio, L.M.; Collado-Arenal, A.M.; Romero-Puertas, M.C. Deciphering peroxisomal reactive species interactome and redox signalling networks. Free Radic. Biol. Med. 2023, 197, 58-70, doi:10.1016/j.freeradbiomed.2023.01.014.
  5. Zandalinas, S.I.; Mittler, R. ROS-induced ROS release in plant and animal cells. Free Radic. Biol. Med. 2018, 122, 21-27, doi:10.1016/j.freeradbiomed.2017.11.028.
  6. Pecchillo Cimmino, T.; Ammendola, R.; Cattaneo, F.; Esposito, G. NOX Dependent ROS Generation and Cell Metabolism. Int. J. Mol. Sci. 2023, 24, doi:10.3390/ijms24032086.
  7. Choudhury, S.; Panda, P.; Sahoo, L.; Panda, S.K. Reactive oxygen species signaling in plants under abiotic stress. Plant Signal Behav. 2013, 8, e23681, doi:10.4161/psb.23681.
  8. Sachdev, S.; Ansari, S.A.; Ansari, M.I.; Fujita, M.; Hasanuzzaman, M. Abiotic Stress and Reactive Oxygen Species: Generation, Signaling, and Defense Mechanisms. Antioxidants (Basel) 2021, 10, doi:10.3390/antiox10020277.
  9. Mailloux, R.J. An update on methods and approaches for interrogating mitochondrial reactive oxygen species production. Redox Biol. 2021, 45, 102044, doi:10.1016/j.redox.2021.102044.
  10. Kettle, A.J.; Ashby, L.V.; Winterbourn, C.C.; Dickerhof, N. Superoxide: The enigmatic chemical chameleon in neutrophil biology. Immunol. Rev. 2023, 314, 181-196, doi:10.1111/imr.13183.
  11. Abe, C.; Miyazawa, T.; Miyazawa, T. Current Use of Fenton Reaction in Drugs and Food. Molecules 2022, 27, doi:10.3390/molecules27175451.

Reviewer 2 Report

Dear Authors,

I had a great pleasure to asses a review paper entitled:” Reactive oxygen species: A hinge between plants and humans” which is considered for publication in IJMS journal. The manuscript itself is good written and contain a lot of valid information associated with ROS however it also need some add-ons and corrections. Because of number/complexity of needed improvements I suggest major revision (it will gave time to make it possible according IJMS rules) but I would like to outline that article is very interesting and valid. The list of needed improvements I present below:

1.       Title section

Authors should modified its overtone. Now the title is too general and not précised formulated. I suggest to change title to “Reactive oxygen species: A crosslink between plant and humans eukaryotic cells” or “Reactive oxygen species: A crosslink between plant and humans eukaryotic cells systems”. This change will direct attention to the truly ROS involvement. Firstly ROS are involvement in signal transduction cells.

2.       Abstract section

It is little too short for review paper it should be extended. The abstract must be catchy and outline the content of review. because

3.       Main sections of article

Most of Paragraphs/chapters are well balanced however the chapter 2.5 Pathogen defense is too short in comparison to other chapters in plants. I suggest add the involvement of ROS for example in cell wall remodeling in reaction for pathogens and plant viruses. This will be outstanding add on. This will also outline the main difference between plant and human cell which is of course presence of plant cell wall. AND ROS are crucial for cell wall. Ther fore I added list of potential publication which could be used for that improvement. The good examples of this type remodeling pathogen could be source from: Lionetti et al 2017 (https://doi.org/10.1104/pp.16.01185), Rigano et al., (https://doi.org/10.1016/j.plantsci.2017.10.013), Torres et al., 2006 (10.1104/pp.106.079467)

 in case of plant viruses it could be found PVY in IJMS (10.3390/ijms19030862), Frontiers in Microbiology (10.3389/fmicb.2021.656809/full) or other publication which could be added in case of other plant viruses

Figure 1 and 2 must be much bigger now are too small. Moreover, the Figure captions must be self-descriptive in review papers. Authors must describe all elements on figure for example add numbers in to each cell reactions/stage and describe all stages in order of this numbers. The only one sentence as Figure descriptions is too little for review.

4.       References-minor problem

The list should be check the abbreviation of journal should be in ISO standard it need to be corrected.

Sincerely,

Author Response

Dear reviewer:

Thank you for your comments on our manuscript entitled “Reactive oxygen species: A hinge between plants and humans” (Manuscript ID:  ijms-2555884). Those comments were very helpful for improving our manuscript. We believed we have answered reviewer’ s all comments. Based on your recommendations, we have made further modifications to the article, particularly in “abstract”, “Pathogen Defense” and Figure descriptions. Below the comments of the reviewer are responded point by point and the revisions article are indicated by red.

Reviewer 2:

I had a great pleasure to asses a review paper entitled:” Reactive oxygen species: A hinge between plants and humans” which is considered for publication in IJMS journal. The manuscript itself is good written and contain a lot of valid information associated with ROS however it also need some add-ons and corrections. Because of number/complexity of needed improvements I suggest major revision (it will gave time to make it possible according IJMS rules) but I would like to outline that article is very interesting and valid. The list of needed improvements I present below:

Comment 1: Title section

Authors should modified its overtone. Now the title is too general and not précised formulated. I suggest to change title to “Reactive oxygen species: A crosslink between plant and humans eukaryotic cells” or “Reactive oxygen species: A crosslink between plant and humans eukaryotic cells systems”. This change will direct attention to the truly ROS involvement. Firstly ROS are involvement in signal transduction cells.

Reply 1: I appreciate your thoughtful feedback regarding the title of the article. Your insights have indeed prompted us to reconsider the clarity and precision of the title. We have modified the title to “Reactive oxygen species: A crosslink between plants and humans eukaryotic cells systems” after consideration to emphasize the pivotal role of ROS in facilitating cross-species cellular interactions and signal transduction pathways.

Comment 2: Abstract section

It is little too short for review paper it should be extended. The abstract must be catchy and outline the content of review. because

 Reply 2: Thank you for your suggestions. We rewrite the abstract as follows.

Reactive oxygen species (ROS) are important regulating factors that play a dual role in plant and human cells. As the first messenger response in organisms, ROS coordinate signals in growth, development, and metabolic activity pathways. They also can act as an alarm mechanism, triggering cellular responses to harmful stimuli. However, excess ROS cause oxidative stress-related damage, oxidize organic substance, leading to cellular malfunctions. This review summarizes the current research status and mechanisms of ROS in plant and human eukaryotic cells, highlighting the differences and similarities between the two and elucidating their interactions with other reactive substances and ROS. Based on the similar regulatory and metabolic ROS pathways in the two kingdoms, this review proposes future developments that can provide opportunities to develop novel strategies for treating human diseases or creating greater agricultural value.

Comment 3: Main sections of article

Most of Paragraphs/chapters are well balanced however the chapter 2.5 Pathogen defense is too short in comparison to other chapters in plants. I suggest add the involvement of ROS for example in cell wall remodeling in reaction for pathogens and plant viruses. This will be outstanding add on. This will also outline the main difference between plant and human cell which is of course presence of plant cell wall. AND ROS are crucial for cell wall. Ther fore I added list of potential publication which could be used for that improvement. The good examples of this type remodeling pathogen could be source from: Lionetti et al 2017 (https://doi.org/10.1104/pp.16.01185), Rigano et al., (https://doi.org/10.1016/j.plantsci.2017.10.013), Torres et al., 2006 (10.1104/pp.106.079467)

 in case of plant viruses it could be found PVY in IJMS (10.3390/ijms19030862), Frontiers in Microbiology (10.3389/fmicb.2021.656809/full) or other publication which could be added in case of other plant viruses

Figure 1 and 2 must be much bigger now are too small. Moreover, the Figure captions must be self-descriptive in review papers. Authors must describe all elements on figure for example add numbers in to each cell reactions/stage and describe all stages in order of this numbers. The only one sentence as Figure descriptions is too little for review.

Reply 3: Thanks for pointing out the inadequacy of the article, after referring to the literature we have added the context of the involvement of ROS in cell wall remodeling in reaction for pathogens and plant viruses from L245 to L257 which are indicated by red.

For the figures’ problem, we have added the guide letters to enhance readability and enlarged the figures for reading. And we expanded the context of figure descriptions in L426 to L428 and L732 to L734 which are indicated by red.

The rewrite text is under blow:

The primary distinguishing factor between plant and human cells is the presence of the plant cell wall. This wall serves as a crucial physical barrier that pathogens must navigate when colonizing plant tissues. Disruptions to cell wall integrity can trigger targeted defense responses by affecting proteins engaged in cell wall biosynthesis, degradation, and remodeling. Such disruptions can arise from cell wall damage caused by both biotic and abiotic stresses. Plants employ inducible defense mechanisms that involve biochemical alterations of cell wall components, shielding them from enzymatic deterioration. Notably, ROS play a pivotal role in orchestrating cell wall remodeling in reaction to pathogens and plant viruses [1,2]. ROS can act as antimicrobial molecules involved in reinforcing plant cell walls and inducing callus deposition to limit pathogen entry [3]. The enzymatic activities of peroxidases [4] and pectin methylesterases [5], influenced by ROS, contribute to the restructuring of cell wall polymers, offering a dynamic response to the evolving threat.

Figure 1. External factors generate ROS directly/indirectly (e.g., heavy metals, UV) in plants, which then regulate functions via CDKs/MAPKs affecting transcription and cell cycle, and control cell death via calcium, mitochondria, and chloroplasts, with ROS waves activating defense.

Figure 2. ROS act as messengers to affect regulating pathways and control immunity, triggered by external factors. Meanwhile mitochondrial issues cause ROS leakage, leading to diseases and apoptosis.

Comment 4:  References-minor problem

The list should be check the abbreviation of journal should be in ISO standard it need to be corrected.

Reply 4: Thank you very much for pointing out the mistakes of references, we have corrected the format of journal in ISO standard by endnote and checked carefully.

Sincerely,

  1. Rigano, M.M.; Lionetti, V.; Raiola, A.; Bellincampi, D.; Barone, A. Pectic enzymes as potential enhancers of ascorbic acid production through the D-galacturonate pathway in Solanaceae. Plant Sci. 2018, 266, 55-63, doi:10.1016/j.plantsci.2017.10.013.
  2. Lionetti, V.; Fabri, E.; De Caroli, M.; Hansen, A.R.; Willats, W.G.; Piro, G.; Bellincampi, D. Three Pectin Methylesterase Inhibitors Protect Cell Wall Integrity for Arabidopsis Immunity to Botrytis. Plant Physiol. 2017, 173, 1844-1863, doi:10.1104/pp.16.01185.
  3. Camejo, D.; Guzman-Cedeno, A.; Moreno, A. Reactive oxygen species, essential molecules, during plant-pathogen interactions. Plant Physiol. Biochem. 2016, 103, 10-23, doi:10.1016/j.plaphy.2016.02.035.
  4. Fujita, S.; De Bellis, D.; Edel, K.H.; Koster, P.; Andersen, T.G.; Schmid-Siegert, E.; Denervaud Tendon, V.; Pfister, A.; Marhavy, P.; Ursache, R.; et al. SCHENGEN receptor module drives localized ROS production and lignification in plant roots. EMBO J. 2020, 39, e103894, doi:10.15252/embj.2019103894.
  5. Li, T.; Shi, D.; Wu, Q.; Yin, C.; Li, F.; Shan, Y.; Duan, X.; Jiang, Y. Mechanism of Cell Wall Polysaccharides Modification in Harvested 'Shatangju' Mandarin (Citrus reticulate Blanco) Fruit Caused by Penicillium italicum. Biomolecules 2019, 9, doi:10.3390/biom9040160.

Round 2

Reviewer 1 Report

The manuscript has been revised accordingly and I don't have further questions.

Author Response

Dear Reviewer

Thank you again for your comments on our manuscript entitled “Reactive oxygen species: A hinge between plants and humans” (Manuscript ID:  ijms-2555884). We gratefully thank you for your time spend making your constructive remarks and useful suggestions, which has significantly raised the quality of the manuscript and has enabled us to improve the manuscript.

Sincerely

Reviewer 2 Report

Dear Authors,

The citation about plant viruses is not correct used. the 77 and 78 did not present any data about plant viruses at all. It must be corrected. I recommend the add appropriate citation as was suggested in my first review: “… in case of plant viruses it could be found PVY in IJMS (10.3390/ijms19030862), Frontiers in Microbiology (10.3389/fmicb.2021.656809/full) or other publication which could be added in case of other plant viruses.” Moreover, also It also well known the role of RBOHD, robhF, or rbohd/f mutants of Arabidopsis and reactive species or even glutathione modulation in to the text of responses to TuMV plant virus the papers was published in IJMS in 2023 (10.3390/ijms24087128) which will be great addiction to too this review especially that authors mention the role of RBOHD in Arabidopsis in lines 263-268 in pathogen response.

Sincerely,

Author Response

Dear reviewer:

Thank you for your comments on our manuscript entitled “Reactive oxygen species: A hinge between plants and humans” (Manuscript ID: ijms2555884). Those comments were very helpful for improving our manuscript. Below, the comments of the reviewer are responded, and the revised parts are indicated by blue.

Reviewer 2:

Comment: The citation about plant viruses is not correct used. the 77 and 78 did not present any data about plant viruses at all. It must be corrected. I recommend the add appropriate citation as was suggested in my first review: “… in case of plant viruses it could be found PVY in IJMS (10.3390/ijms19030862), Frontiers in Microbiology (10.3389/fmicb.2021.656809/full) or other publication which could be added in case of other plant viruses.” Moreover, also It also well known the role of RBOHD, robhF, or rbohd/f mutants of Arabidopsis and reactive species or even glutathione modulation in to the text of responses to TuMV plant virus the papers was published in IJMS in 2023 (10.3390/ijms24087128) which will be great addiction to too this review especially that authors mention the role of RBOHD in Arabidopsis in lines 263-268 in pathogen response.

Reply: Thank you for addressing the missing references in our main text concerning plant virus-related literature and the significance of glutathione and the rboh family in the context of TuMV infection in plants. We have successfully added the necessary references (79, 80) in L258 and enriched this section of the main text from L262 to L264.

The revised text is under below:

In the Arabidopsis–TuMV pathosystem response, glutathione can act as a key signaling factor in not only susceptible rbohD reaction but also the resistance reaction presented by rbohF and rbohD/F mutants [1].

  1. Otulak-Koziel, K.; Koziel, E.; Treder, K.; Kiraly, L. Glutathione Contribution in Interactions between Turnip mosaic virus and Arabidopsis thaliana Mutants Lacking Respiratory Burst Oxidase Homologs D and F. Int. J. Mol. Sci. 2023, 24, doi:10.3390/ijms24087128.

Sincerely

Round 3

Reviewer 2 Report

Dear Authors,

All improvements was added. I recommend publication. Very good review paper.

Sincerely,